# Pregnancy Rates of Holstein Friesian Cows with Cavitary or Compact Corpus Luteum

**DOI:** 10.3390/vetsci11060246

**Published:** 2024-05-30

**Authors:** Boglárka Vincze, Levente Kátai, Kamilla Deák, Krisztina Nagy, Sándor Cseh, Levente Kovács

**Affiliations:** 1Embryo Transfer Station, Department of Obstetrics and Production Animal Medicine Clinic, University of Veterinary Medicine, István utca 2, 1078 Budapest, Hungary; vincze.boglarka@univet.hu (B.V.); deak.kamilla@gmail.com (K.D.); cseh.sandor@univet.hu (S.C.); 2EmbrioVet Ltd., Vilmos utca 13, 7625 Pécs, Hungary; info@embriovet.hu; 3Institute of Animal Sciences, Hungarian University of Agriculture and Life Sciences, Páter Károly utca 1, 2100 Gödöllő, Hungary; kris.nagy@googlemail.com

**Keywords:** cavitary, cystic ovarian disease, ovary, pregnancy rate, milk yield, dairy cows

## Abstract

**Simple Summary:**

From the viewpoint of production, reproduction is very important in the life of a dairy farm cattle since there is no production without reproduction. This manuscript describes whether the presence of a cavity in the corpus luteum 31–42 days after breeding may have an influence on pregnancy outcomes in dairy cattle. Several observations have been made speculating about whether cavitary corpora lutea limit pregnancy success, but only a few quantitative assessments and no large-scale evaluations have been made to prove the risk of a possible reduced pregnancy rate. After evaluating a large number of cases/observations, our study concludes that cavitary corpora lutea is considered an ‘enemy’ for open cows.

**Abstract:**

Cavitary corpora lutea are commonly observed during the estrous cycle in bovines. Since the quality of the corpus luteum (CL) is extremely important during embryo transfer when embryos are implanted into the recipient, the ultrasonographic examination of the CL is becoming more and more important in terms of the outcome of the procedure. In the present study, a total of 2477 ultrasonographic transrectal diagnoses were performed, and data were collected between the years of 2018 and 2020 in a large-scale Holstein Friesian dairy farm in Hungary. In 91.1% (n = 2257) and in 8.9% (n = 220) of the cases, compact CLs and cavitary CLs, respectively, were diagnosed at pregnancy diagnosis. The presence of a cavitary CL on the ovary at pregnancy diagnosis increased the odds of remaining open after pregnancy by 21 times compared to the presence of a compact CL (OR = 21.0, *p* < 0.001) in the cows. The presence of cavitary CL was not influenced either by month or season. Ovarian cysts were detected in 196 cases (8.0%) in the examined animals. The presence of a cavitary CL decreased by 9 times when an ovarian cyst was also diagnosed (OR = 9.0, 1.6% vs. 9.5%, *p* < 0.001). The presence of an ovarian cyst decreased the odds of established pregnancy by 81 times (OR = 81.1, *p* < 0.001). Based on our results, the presence of a cavitary CL between days 31 and 42 after artificial insemination is associated with a smaller chance of conception in Holstein Friesian cows. The presence of an ovarian cyst decreases the occurrence of cavitary CL and the chance of conception.

## 1. Introduction

Practitioners working in the field quite often find different ‘abnormal’ fluid-filled ovarian structures on cows’ ovaries during their routine work [1]. Diagnosis can be challenging, because different and often confusing definitions and descriptions are used in the literature for these structures; as a summarizing term, these structures are often mentioned as cystic ovarian disease (COD) [1,2,3]. These structures may be anovulatory or ovulatory in origin [3], the latter sometimes considered as a cavitary corpus luteum (CL), which can remain as it is, or the cavity can disappear during the cycle [4,5,6,7]. The occurrence of cystic structures and cavitary corpora lutea (CLs) were investigated mostly during the 1980s by means of transrectal palpation or slaughterhouse organ collection. Years later, transrectal ultrasonographic (TRUS) studies revealed some facts about the development and nature of cavitary CLs on bovine ovaries [8,9].

A central cavity within the bovine CL is commonly observed during the estrous cycle and early pregnancy via TRUS [2,4,6,8]. This cavity contains mostly clear, serous fluid and sometimes fibrin strains [8]. The incidence of the cavity varies between 25–78% in different sources [8,9,10,11]. This cavity reaches its maximum diameter during days 6–9 of the estrous cycle and remain or disappear later during the cycle [9], with larger cavities disappearing later in the cycle showing no significant difference in progesterone production compared to compact CLs. Cavitary CLs can have different-sized lumen varying between <2 mm and more than 10 mm when they are present [8]. The types of CLs have been studied in several ways previously, but contradictory results arise in the question of whether the different types of CLs could influence pregnancy outcomes. Information has been gathered on a possible difference between the progesterone production of cavitary CLs and compact ones; embryo transfer studies investigated whether there is any difference in outcome related to the type of CL, but no large-scale examinations have been conducted to investigate the association between cavitary CL and pregnancy outcome in a practical setting. This study aims to show the negative effects of the presence of a cavitary CL on pregnancy outcome in dairy cows, which has not been investigated so far.

Recipients with CLs with a cavity in the center are also usually considered ‘unsuitable’ for embryo transfer (ET) or they are thought to be ‘less efficient’ for maintaining pregnancy in cattle [11]. However, the basis of such negative judgements has not been clearly proven to date. Although cavitary CLs are still considered to be a physiological state or type of CL, clinical experience indicates that prolonged interovulatory intervals and early embryonic loss can be associated with the appearance of these structures [8,9,10,11,12,13,14]. Some studies investigating the function of cavitary, and compact CLs have shown contradictory results so far [11,12,13,14,15].

The aim of this study was to investigate the relationship between the type of CL (cavitary or compact) visible on the day of pregnancy diagnosis, pregnancy rates, and some other parameters (parity, the presence of cavitary CL, the presence of a cyst, and season) evaluated in a large-scale commercial dairy farm from a practical point of view.

## 2. Materials and Methods

### 2.1. Ethical Authorization of Animal Experiment

All methods and the procedures applied on the animals were carried out in accordance with the European Union Directive of the European Parliament and the Council on the protection of animals assessed for scientific purposes. Study permit number: PE/EA/1973-6/2016 (issued by Pest County Government Office, Department of Animal Health).

### 2.2. Housing, Data Collection, and Reproductive Management of Animals

Data were collected during the years of 2018–2020 on a large-scale commercial dairy farm with a herd of over 1000 lactating cows, located in northern central Hungary, Pest County. A total of 1128 asymptomatic [without any obvious disease, i.e., mastitis, lameness, endometritis, etc. at the time of artificial insemination (AI) or TRUS] lactating Holstein Friesian cows [parity: 3.4 (range 1–8); age: 6.3 (range 2–10) years; daily milk yield: 42.3 (range 28–53 kg); body condition score: 3.5 (range 3.25–3.75); days in milk: 152 (78–241 days)] were enrolled in the study. The distribution of parity of cows is shown in Figure 1. A total of 2477 artificial inseminations (AI) and transrectal ultrasonographic pregnancy examinations were analyzed using the data of 1128 cows. The date of pregnancy examination, parity, last calving date, date of the last AI, number of services, milk yield at the date of the TRUS, days in milk, and TRUS findings on both ovaries and the uterus were used for the analysis.

Cows were housed in free stall barns with ad libitum access to feed and water and were fed with total mixed ration two times daily, which followed the NRC guidelines for high-producing dairy cows [16]. Cows were milked twice daily at approximately 12 h intervals (morning milking 06:00–09:00, evening milking 17:00–20:00 CET+1), and average 7-day daily milk production was recorded by the farm management system software (RISKA 5.2.8. software, Hungary). Animals were under strict veterinary control during the study period. During the experimental period, a 70-day voluntary waiting period (VWP) was given to cows after calving. After this period, cows were presented to a veterinary reproduction control specialist with ultrasound scanning to detect ovarian cyclicity. For estrus synchronization and AI, fixed-time AI (FTAI) programs were applied (OvSynch and Re-Synch) with GnRH-analogue and prostaglandin-analogue cloprostenol (G-P-G) as follows: Day 1 GnRH inj., Day 7 prostaglandin inj., Day 9 GnRH inj., Day 10 AI (OvSynch FTAI); and Day 1 prostaglandin inj., Day 3 GnRH inj., Day 4 AI (Re-Synch FTAI program) [16]. Cows were inseminated with commercially available frozen-thawed semen from proven fertile bulls. Before AI, cows were checked for clinical signs of estrus and the presence of clear vaginal mucus. On this farm, the identification of animals was performed with an on-cow accelerometer (Heatime, SCR Engineers Ltd., Netanya, Israel) attached to a neck collar placed 3 days after calving when the animal left the postpartum stall unit. Only estrous and healthy (free from reproductive or obvious clinical disorders) cows were inseminated by a well-trained technician.

### 2.3. Transrectal Ultrasound Examination (TRUS)

Routine transrectal palpation and TRUS for pregnancy diagnosis were performed between Days 31 and 42 after AI with a portable 5 MHz machine with a linear endorectal probe (GE Logiq V2 portable ultrasound device, Med-En-Trade Ltd., Budapest, Hungary), as described and previously validated by [17] and described in detail in Appendix A. The number and the viability of embryos (if any) in the uterus and the presence and type of the CL and other ovarian structures were recorded. Uterine horns were scanned using a transrectal approach upon the dorsal surface continuously multiple times, looking for the yolk sac, embryonic fluid, the body of the embryo, and the embryonic heart and heart beats. If the cow was diagnosed as non-pregnant and a CL was present on one of the ovaries, a Re-Synch program was applied on the same day, as described earlier. Twin pregnancies were also recorded, and the viability of each embryo was checked, as described earlier.

Both ovaries were scanned for the presence of ovarian structures and the number and type of CLs (Figure 1). A cavitary or compact CL was identified by its ultrasonographic appearance (granular, grey structured area in the ovarian tissue), as described by Kähn [6]. The CL is hypogenic compared to ovarian stroma, with a brighter appearance; it can be recognized by the weaker echogenicity of its luteal tissue [6]. Corpora lutea with cavities (also referred to as cystic CLs) have an echoic rim of tissue (a few millimeters thick, described by Kähn [6,18]) that surrounds a central anechoic fluid accumulation. Cavitary CLs have thus been recognized by the above categorization previously defined by Kähn [6]. CLs could be distinguished from luteinized cysts by the ovulation papilla that should distort the outline of the cyst at the point of ovulation (if visible) [19]. Ovarian structures other than follicles (e.g., cysts) were recorded (Figure 1 and diagnosed as follows: the structure was considered as a cyst if it was a fluid-filled structure that was at least 20 mm in size without the presence of a CL [1,19]. Ultrasonographically, the two types of ovarian cysts (thecal follicular cyst and luteinized follicular cyst) could be distinguished using the previous description by Kähn; a large-sized thin-walled spherical-shaped cyst; or, in the latter case, a thick-walled large-sized cyst with a hypoechoic wall, similar to that in CLs with luteal tissue [6]. Upon data collection, no difference was found between follicular (thin-walled, less than 3 mm wall thickness as defined) and luteal (luteinized wall with more than 3 mm wall thickness) cysts [19].

### 2.4. Statistical Analysis

All analyses were carried out using R Statistical Software 4.1.3 [20]. The significance level was set at *p* < 0.05. A binomial test and corresponding 95% confidence interval (CI) were used to test whether the occurrence of twin pregnancies differs significantly from the value reported in the previous literature, known to be 5%, and to determine whether there is a preference for left or right side regarding the position of the embryo or the cavitary CL. Twins were excluded from the pregnancy site analysis because the actual site of the pregnant horn was not always verified when twins occupied the different areas of uterine horns.

A mixed-effect multivariate logistic regression model with random effects for each cow was used to determine which of the examined factors were the best predictors of the presence of pregnancy (model 1), and the presence of cavitary CL (model 2) using backward elimination. To compare milk production between cows, a general linear mixed model was fit to the data (model 3). Normality was checked for milk production by using a Q–Q plot, and the results suggested no apparent violation of the assumption. In each model, the experimental unit was the pregnancy diagnosis, and cows served as random effects to account for the fact that cows were measured several times. The fixed effects for models 1, 2, and 3 were parity (one or more), the presence of a cyst (yes or no), season (June, July, and August were considered as the summer season, while the rest of the months were categorized as non-summer seasons), and number of services (1–3 or more than 3). For models 1 and 3, the presence of cavitary CL (yes or no) was added to the fixed effects as well.

The penalized quasi likelihood method was used to estimate the parameters in models 1 and 2 [21,22]. The removal of the non-significant factors resulted in models with a lower Akaike information criterion in each case, interpreted as the explanatory power of the initial and final models being the same. The exponentials of b-coefficients in the final models in models 1 and 2 were interpreted as odds ratios of the outcome variable.

## 3. Results

A total of 2397 pregnancy diagnoses were performed and analyzed during the study period using the data of 1128 cows retrospectively. More than one-third (37.5%; n = 898) of the AIs were performed on primiparous cows and 62.5% (n = 1499) were performed on multiparous cows. In 28 cases (1.1%), ultrasonography-visible embryonic/early fetal mortality cases (cloudy embryonic fluid/yolk sac, absence of heartbeat, other signs of dead embryo or membrane remnants) were diagnosed, which were considered as pregnancy loss and counted as non-pregnant cases. A total of 964 cows (40.2%) became pregnant; out of these cows, 72.2% (n = 696) of them had services between 1 and 3 per conception, and 27.8% (n = 268) of them had 4 or more services prior to pregnancy. There were 56 (5.9%) twin pregnancies diagnosed, which does not differ significantly from the previously observed 5% (*p* = 0.237, 95% CI: 0.044–0.075).

Not considering the data on twin pregnancies (n = 908), significantly more embryos were on the right uterine horn compared to the left uterine horn (57.9%, n = 525 vs. 42.1%, n = 382; *p* < 0.001, 95% CI: 0.55–0.61). Cavitary CLs were diagnosed 216 times (9.2%) with TRUS, and 90.8% of cows (n = 2124) had compact CL. The results of model 1 highlighted that the odds of positive pregnancy diagnosis were different among cows, as shown in Table 1.

The results of model 2 highlighted that the odds of positive cavitary CL diagnosis were not influenced significantly by parity (*p* = 0.168). The results of model 2 are shown in Table 1. Monthly frequencies of pregnancy and cavitary CL diagnoses during the study is shown by Figure 2. The results of model 3 highlighted that milk production was not influenced significantly by the presence of a cavitary CL, ovarian cyst, or the season variable (*p* > 0.1 in all cases). However, multiparous cows had significantly higher daily milk production on the week of the diagnosis compared to primiparous cows (44.1 ± 7.3 kg vs. 38.1 ± 5.9 kg, F = 6.5, *p* < 0.001, Figure 3). Also, cows during the 1st–3rd services had higher daily milk production on the week of the diagnosis compared to the cows examined after the 3rd service (43.1 ± 8.3 kg vs. 38.4 ± 7.3 kg, F = 4.8, *p* < 0.001).

## 4. Discussion

In this study, more conceptuses were found in the right uterine horn, which coincides with previous findings of the distribution of conceptuses within the left and right uterine horns [23,24]. The 1.1% embryonic mortality rate found here differs from expectation, as the actual embryonic/early fetal mortality rate had been thought to be much higher. This may be because cases cannot be identified when they occur between the time of AI and the TRUS examination and therefore remain undetected. Our results reflected pregnancy losses that were seen at the exact time of the TRUS and not earlier or later. The twin pregnancy rate in this study agreed with earlier findings, which reported a 5.7% prevalence of twin pregnancy occurrence during an 11-year period in Holstein Friesian large-scale farms [25].

Primiparous cows were overrepresented (72%) in this study; the odds ratio for a positive pregnancy diagnosis was 1.9 compared to multiparous cows, which reflects literature data of higher conception rates reported in most of the herds (31–39% for primiparous cows and 24% or less for multiparous cows) [26,27]; however, conception rate is influenced excessively by several other conditions (e.g., season, differences in reproductive management protocols, experience of the operator, etc.), which makes the comparability of the published results quite difficult [26]. Conception rate or conception risk is calculated from early pregnancy diagnoses, so the actual physiological fertilization rate is thought to be above 80%; however, fertilization does not seem to be the principal factor responsible for the low fertility in single-ovulating cows [28].

From the 1970s, several definitions have been created to describe different cavitary structures such as follicular cysts, cystic Graafian follicles, luteinized follicles, luteinized cysts, cystic CL, and vacuolized CL [1,6,18,29]. Nowadays, the most accepted definition of an ovarian cyst is “a fluid-filled follicle, with a diameter at least 20 mm present on one or both ovaries and persists at least 10 days on the ovary without the presence of active luteal tissue” [1,19]. By means of TRUS, central cavities in the bovine CL are first detectable on days 3 to 5 of the estrous cycle and proven to reach maximum diameters on days 6 to 12 [8,9,10,30,31]. Thereafter, most cavities decrease in size and disappear on days 16 to 20 [8,9,31], although some large cavities persist until days 21 to 48 after AI [9]. As described previously [31], only a small number of cavities (7%) can be still detected between days 30 and 42, when pregnancy examinations were performed in this study and 8.9% of the CLs were detected with cavity. A limitation of this study is that a higher percent of cavitary CL was suspected, but most of them may have disappeared by the time of TRUS. However, the aim of the study was to evaluate the CLs that remained cavitary during early gestation and were associated with a positive or negative pregnancy status.

The major finding of our study is a higher pregnancy rate of 43.4% (900/2124) in cases of compact CL, while it was only 3.7% in cows detected with cavitary CL at pregnancy diagnosis. Hence, the presence of a cavitary CL increased the odds of remaining open after service by 33.8 times compared to compact CL in the examined cases. Several studies have demonstrated that the ability of the CL to produce P4 and maintain pregnancy depends on the morphology (cavitary or compact). Barreiros et al. [32] found 22.7% cavitary CL in crossbred recipients when the animals were synchronized with progesterone. Another aspect observed by Kastelic et al. [10] was the loss of the cavities of CL in pregnant heifers. Such a loss of cavity detected by TRUS examination demonstrated that the maintenance of the early pregnancy is independent of the presence or absence of the cavity. Additionally, other studies showed no difference in plasma P4 concentrations in dairy cows with cavitary or compact CL at the time of ET [32,33].

As a relatively common finding in dairy cows, especially in the postpartum period, ovarian cysts are associated with subfertility [34]. The pregnancy rate presented here was nearly 50% in non-cystic cows, while it was only 1.1% in the presence of at least one ovarian cyst. Hence, the presence of an ovarian cyst decreased the odds of an established pregnancy by 95 times, and cows with at least one cyst produced more milk than the others. This agrees with previous studies that reported a strong association between milk yield and the presence of cystic ovarian structures [35,36]. With the intensive selection for milk production, COD has become an everyday problem for bovine practitioners worldwide; hormonal disturbances, increased days open, and increased veterinary costs cause significant economic losses associated with cystic ovarian disease.

Cavitary CLs were diagnosed more frequently between 1st June until 31st August than in other parts of the year (Table 1). No studies have evaluated the frequency of cavitary CLs by season so far. Milk production was not influenced by season nor by the type of CL. Although no specific meteorological measurements were taken during the study, this result is surprising, as lower milk yields are usually reported in dairy cattle studies during elevated ambient temperatures in summer [37,38,39,40].

Our results suggest that unknown causatives might lead to the development of a cavitary CL and to the loss of the conceptus. It has been reported that a very large cavity and thin luteinized tissue formation in the CL results in fertility reduction [41]. Therefore, the size of the CL luteal area (smaller when there is a central cavity) may serve as a diagnostic marker for the detection of non-pregnant cattle around the luteal regression phase [42]. When it comes to pregnancy loss prediction, the size or diameter of the CL, regardless to its type, does not seem to be a good predictor [43,44]. However, the size of the peri-ovulatory follicle may predict pregnancy failure [45]. In Doppler ultrasonography studies, the blood flow of different tissues was studied to identify a possible relationship with the altered progesterone-producing activity of the CL tissue, and a positive correlation between the circulating progesterone concentration and CL blood flow was proven up to day 40 of gestation [46]. Blood perfusion rates of the CL are also reliable predictors of pregnancy status from days 18 to 24 in both beef and dairy cattle, presumably because these CLs survive the maternal recognition of the early pregnancy [13,47].

## 5. Conclusions

Based on the results presented here, a cavitary CL diagnosed between days 31 and 42 after AI associates with a smaller chance of conception in Holstein Friesian cows. Factors possibly responsible for the formation of a cavitary CL might lead to non-pregnancy in the affected animals. The presence of a compact CL does not lead to successful pregnancy in all cases. The presence of an ovarian cyst decreases the odds of carrying a cavitary CL and the chance of conception. Ovarian cysts seem to be associated with a higher milk production at the time of pregnancy diagnosis; however, the exact reasons of this phenomenon are not yet known.

## Figures and Tables

**Figure 1 vetsci-11-00246-f001:**
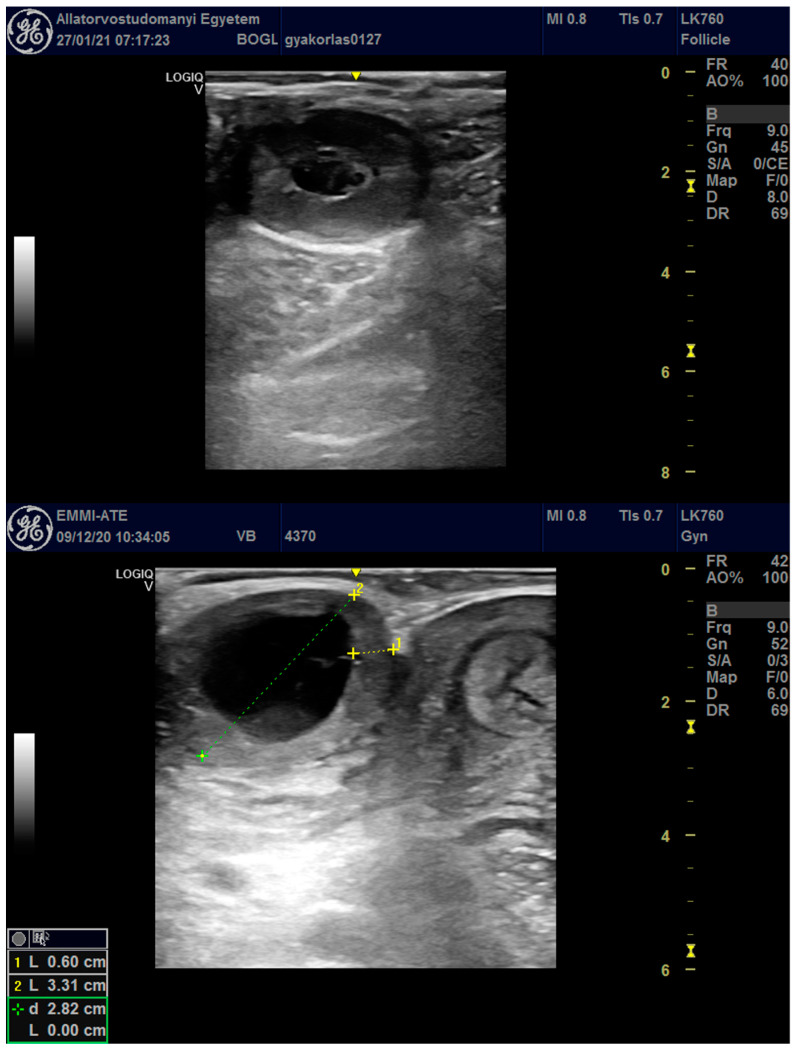
A cavitary CL (**up**) and a luteinized follicular cyst with a luteinized wall and a large cavity in the center (**down**). The diameter (depicted with calipers on the image) and the form indicate its suspected origin, but with the ultrasound alone, the diagnosis is not definitive. Both structures are scanned with a 5 MHz endorectal probe (GE Logiq V2 ultrasound device).

**Figure 2 vetsci-11-00246-f002:**
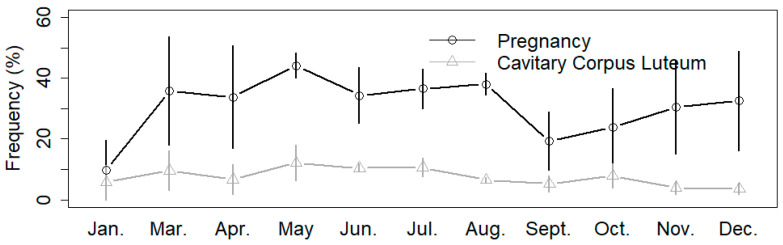
Monthly frequencies of pregnancy and cavitary CL diagnoses during the 3-year study period.

**Figure 3 vetsci-11-00246-f003:**
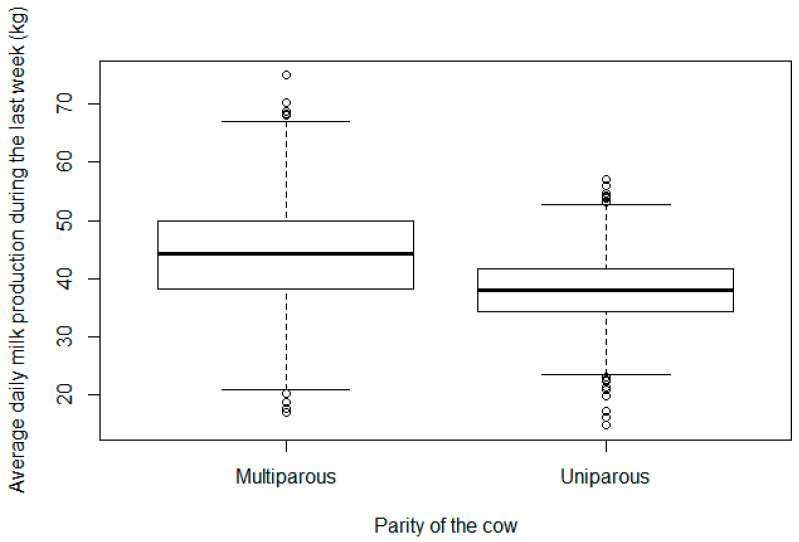
Average daily milk production by parity on the week of the diagnosis (kg).

**Table 1 vetsci-11-00246-t001:** Pregnancy rates and their odds ratios (ORs) in the different groups of cows.

Variables	Odds Ratio (OR)	Significancy	Prevalence
Pregnancy of primiparous vs. multiparous cows	1.9	*p* < 0.001	46.3% vs. 34.4%
Pregnant with compact CL	33.8	*p* < 0.001	3.7% vs. 43.4%
Pregnant without ovarian cyst	95.0	*p* < 0.001	1.1% vs. 41.9%
Diagnosed pregnant between 1 June and 31 August	0.6	*p* < 0.001	33.6% vs. 40.9%
Compact CL if ovarian cyst diagnosed	29.4	*p* < 0.001	1.7% vs. 9.9%
Cavitary CL if inseminated between 1 June and 31 August	1.4	*p* = 0.004	9.4% vs. 9.2%
Presence of cavitary CL after 1–3 AI	3.3	*p* < 0.001	10.2% vs. 6.4%

## Data Availability

All data contained within the article and Appendix A.

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
