# Peer review of "Pregnancy Rates of Holstein Friesian Cows with Cavitary or Compact Corpus Luteum"

_vetsci, 2024, doi:10.3390/vetsci11060246_

Round 1

Reviewer 1 Report

Comments and Suggestions for Authors

CRITIQUE This is an observational study which tested the hypothesis that the cavitary CL may be associated with reduced pregnancy rates compared to compact CL. The main limitations are that authors did not measure the size (area or volume of the CL) and did not measure progesterone peripheral/milk concentrations. Also, the practical application is limited, because the assessment of the CL type can only be performed at pregnancy diagnosis: thus, if a cow is pregnant with a cavitary CL, what should/could the vet do to improve the survival of such pregnancy?

On the contrary if the cow is open, she has to be inseminated again, regardless of the type of the CL, so what would be the rationale to assess the CL type?

ABSTRACT, L29: please define when CL assessment (cavitary vs compact) was performed. Was it at pregnancy diagnosis?

L31: The sentence “the presence of a cavitary CL…P<0.001)” is unclear, please revise.

L37: “When an ovarian cyst was also diagnosed.”

INTRODUCTION, page 2 L72: I suggest authors replace “some other parameters” with the list of parameters hat were evaluated.

MATERIALS AND METHODS, L78: if available, please insert the number of the ethical authorisation.

L156 Milk production should be included in the first statistical model to clarify whereas it influenced the likelihood of pregnancy among producing cows.

RESULTS L183: I presume that that by cases authors mean pregnancies and thus open cows were excluded from this analysis. If so, please replace cases with pregnancies.

L191 This result is previously reported in L 182, please delete.

Pregnancy rates and their odds should be presented in a table.

L200 Was milk production higher or lower in cows with cavitary vd compact CL? Please, clarify.

Comments on the Quality of English Language

English revision is recommended.

Author Response

Dear Reviewer,

First of all, we would like to thank you for your time and effort to improve the quality of our manuscript.

We tried to correct everything you suggested and was possible at this time. Below you can read and find the lines (L…) where the modifications have been made in the manuscript and the new manuscript file where you can see the yellow highlighted material including all the changes that were suggested by all 5 reviewers.

Rew#1: This is an observational study which tested the hypothesis that the cavitary CL may be associated with reduced pregnancy rates compared to compact CL. The main limitations are that authors did not measure the size (area or volume of the CL) and did not measure progesterone peripheral/milk concentrations. Also, the practical application is limited, because the assessment of the CL type can only be performed at pregnancy diagnosis: thus, if a cow is pregnant with a cavitary CL, what should/could the vet do to improve the survival of such pregnancy?

On the contrary if the cow is open, she has to be inseminated again, regardless of the type of the CL, so what would be the rationale to assess the CL type?

AU: Although we cannot provide solution for the problem, different management decisions can be made based on the quality of the corpus luteum in a certain cow (whether to transfer an embryo or apply a different hormone protocol for synchronization) if we know the type of the corpus luteum.

Rew#1: ABSTRACT, L29: please define when CL assessment (cavitary vs compact) was performed. Was it at pregnancy diagnosis?

AU: L31: „at pregnancy diagnosis” inserted.

Rew#1: L31: The sentence “the presence of a cavitary CL…P<0.001)” is unclear, please revise.

AU: L32-34 the sentence has been revised.

Rew#1: L37: “When an ovarian cyst was also diagnosed.”

AU: modified (now Line 38).

Rew#1: INTRODUCTION, page 2 L72: I suggest authors replace “some other parameters” with the list of parameters hat were evaluated.

AU: Parameters included by names L79-80

Rew#1: MATERIALS AND METHODS, L78: if available, please insert the number of the ethical authorisation.

AU: Permit number has been inserted here. (L86-88).

Rew#1: L156 Milk production should be included in the first statistical model to clarify whereas it influenced the likelihood of pregnancy among producing cows.

AU: To compare milk production, general linear mixed model was fit to the data (Pinheiro and Bates, 2000) with random effects for each cow. Milk production was not influenced significantly by the presence of a cavitary corpus or the season (general linear mixed model, p>0.5 in both cases).

Rew#1: RESULTS L183: I presume that that by cases authors mean pregnancies and thus open cows were excluded from this analysis. If so, please replace cases with pregnancies.

AU: Term has been changed to „examinations” L207.

Rew#1: L191 This result is previously reported in L 182, please delete.

AU: The double sentence has been deleted.

Rew#1: Pregnancy rates and their odds should be presented in a table.

AU: Now presented as Table.

Rew#1: L200 Was milk production higher or lower in cows with cavitary vd compact CL? Please, clarify.

AU: A new sentence inserted: L234-235:

“Milk production was not influenced significantly by the presence of a cavitary corpus luteum or the season.”

Thank you very much for your expertise-driven suggestions!

Yours faithfully,

the authors

Reviewer 2 Report

Comments and Suggestions for Authors

Comments and questions are included in the text in the comments

Author Response

Dear Reviewer,

First of all, we would like to thank you for your time and effort to improve the quality of our manuscript.

We tried to correct everything you suggested and was possible at this time. Below you can read and find the lines (L…) where the modifications have been made in the manuscript and the new manuscript file where you can see the yellow highlighted material including all the changes that were suggested by all 5 reviewers.

We have modified the manuscript to be more precise upon the decision and materials and methods section as you suggested:

Lines 143-150.

“Corpora lutea are hypogenic compared to ovarian stroma with brighter appearance; it can be recognized upon its weaker echogenicity of luteal tissue [6] . Corpora lutea with cavities (also referred as cystic corpora lutea) have an echoic rim of tissue (a few millimetres thick, described by Kähn [6]) that surrounds a central anechoic fluid accumulation. Cavitary corpora lutea thus have been recognized by the above categorization previously defined by Kähn [6].Cystic corpora lutea could be distinguished from luteinized cysts by the ovulation papilla that should distort the outline of the cyst at the point of ovulation (if visible) [19]

L153-156.

“Ultrasonographically, the two types of ovarian cysts (thecal follicular cyst and luteinized follicular cyst) could be distinguished by previous description of Kähn; large sized thin walled spherical shaped cyst or in the latter case – thick walled large sized cyst with hypoechoic wall similar to that in corpora lutea with luteal tissue [6].”

L164. “Luteinized follicular cyst with a luteinized wall and a large cavity in the centre. The diameter and the form indicate its suspected origin but with the ultrasound alone the diagnosis is not definitive (endorectal probe 5 MHz, GE Logiq V2 ultrasound device)”.

Thank you very much for your expertise-driven suggestions!

Yours faithfully,

the authors

Reviewer 3 Report

Comments and Suggestions for Authors

Authors presented a study regarding the influence of different types of CL on pregnancy outcomes. 

In my opinion, the topic is interesting but is widely discussed in bibliography. Authors collected some retrospective data and tried to compare. However, M&M is not properly described, they do not show the knowledge gap in the introduction section to justify their study, and, in my humble opinion, they did not explode as much as posible the data they collected to show a good result analyses. Moreover, a P4 serum analysis would be really welcome.

I encourage them to keep working on these data prior to publish them.

Some other comments:

- Abstract is too long.

- Use always the same term: corpora lutea or corpus luteum. Different ways throughout the text. Standardize.

- ln 61-62. Super interesenting studies and description. Expand this information regarding different percentages found in bibliography. Maybe due DIM, lactation number, breed, etc.

- ln 65 and so on. use "..........."

- ln 66. reference without format

- Introduction must be eimproved in general to show the importance of this studye, revealing that there is a gap in knowledge about this topic and this study helps to clarify this issue.

- ln 86-88. These data are results. Here you can describe your inclusion criteria but no results.

- Results: missing the description of the study: I guess retrospective.

- ln 97. reference without format

- ln 104-105 missing ref of protocols

- ln 108. sex-shorted?

- figure 2 is not necessary. I would delete it

- figures 3 and 4, although there are excellent, I think they do not give important information to the study. 

- Please make sure you describe properly and with a good reference the difference on wall thickness to differenciate a cyst from a cavitary CL.

- missing P4 serum analysis. It could give super interesting information.

- Statistics: missing normality test. I guess you did a univariate model previously to determine which variables you include in the GLMM, is it correct?? Please explain. Missing correlation procedures.

- ln 174. write 1,027

- ln 178. it is pregnancy but then pregnancy loss...

- ln 179. all with numbers

- ln 179. several is 1 to 3??? 

- ln 182-183 and 192 are giving same results with different percentages. Please review

Comments on the Quality of English Language

minor

Author Response

Dear Reviewer,

First of all, we would like to thank you for your time and effort to improve the quality of our manuscript.

We tried to correct everything you suggested and was possible at this time. Below you can read and find the lines (L…) where the modifications have been made in the manuscript and the new manuscript file where you can see the yellow highlighted material including all the changes that were suggested by all 5 reviewers.

Original review

In my opinion, the topic is interesting but is widely discussed in bibliography. Authors collected some retrospective data and tried to compare. However, M&M is not properly described, they do not show the knowledge gap in the introduction section to justify their study, and, in my humble opinion, they did not explode as much as posible the data they collected to show a good result analyses. Moreover, a P4 serum analysis would be really welcome.

I encourage them to keep working on these data prior to publish them.

AU: Although there have been publications on the possible effect of a cavity in a corpus luteum, cavitary CLs are avoided in embryo transfer programs and clinical experience indicated that very few pregnancies have seen when a cow had cavitary corpus luteum. The previously published observations used small numbers of studied cows.

Unfortunately, we had no possibility and source for P4-measurements in this study and the non-significant difference published previously in hormone levels in cows with compact or cavitary CL has led us to the conclusion that maybe not the P4 level is the parameter that influence the chance of pregnancy in the studied cows.

Some other comments:

- Abstract is too long.

AU: Because the other 4 reviewer did not suggest changing the length, in this round we kept the original length of the Abstract, but if the Editor orders to cut it we definitely will do that if you agree.

- Use always the same term: corpora lutea or corpus luteum. Different ways throughout the text. Standardize.

AU: We standardized throughout the text so only corpus luteum or corpora lutea (in multiple numbers) are written.

- ln 61-62. Super interesting studies and description. Expand this information regarding different percentages found in bibliography. Maybe due DIM, lactation number, breed, etc.

AU: L65-68 contain more information from the cited articles.

- ln 65 and so on. use "..........."

AU: Changed to the internationally used one.

- ln 66. reference without format.

AU: Inserted.

- Introduction must be eimproved in general to show the importance of this studye, revealing that there is a gap in knowledge about this topic and this study helps to clarify this issue.

- ln 86-88. These data are results. Here you can describe your inclusion criteria but no results.

AU: We find the information important to the reader here about the studied population (age, lactational status etc.)

- Results: missing the description of the study: I guess retrospective.

AU: L195 ’retrospective’ inserted.

- ln 97. reference without format

AU: Inserted. (L106 now)

- ln 104-105 missing ref of protocols

AU: Citation for Ovsynch and Resynch has been inserted (Pursley et al 1997.) L117

- ln 108. sex-shorted?

AU: Not sex-sorted, only commercial frozen semen from known fertile bulls.

- figure 2 is not necessary. I would delete it.

- figures 3 and 4, although there are excellent, I think they do not give important information to the study.

AU: To the comment about figures 2-4.: we think this is helpful for the reader to understand what findings we recognized as cavitary CL and ovarian cyst and pregnancy. The other 4 reviewer did not suggest deleting but if the editor decides to remove, we will do.

- Please make sure you describe properly and with a good reference the difference on wall thickness to differenciate a cyst from a cavitary CL.

AU: We have modified the description to a more precise one (Lines 143-150 and 153-156).

- missing P4 serum analysis. It could give super interesting information.

AU: We definitely agree but had no chance to measure P4 from the animals in this large-scale herd. Interestingly previously no differences have been detected in progesterone levels in cavitary and compact CL cows.

- Statistics: missing normality test. I guess you did a univariate model previously to determine which variables you include in the GLMM, is it correct?? Please explain. Missing correlation procedures.

AU: We did normality tests of course and now we provided the missing information. Line 177-178.

- ln 174. write 1,027

AU: Corrected. Line 197. We could not start the sentence with a number.

- ln 178. it is pregnancy but then pregnancy loss...

AU: Pregnancy loss inserted. L202

- ln 179. all with numbers

AU: Unfortunately, we are not allowed to start a sentence with number only with words.

- ln 179. several is 1 to 3??? 

AU: We removed the wrong term. L202-203.

- ln 182-183 and 192 are giving same results with different percentages. Please review

AU: We deleted the double result. L205-213.

Thank you very much for your expertise-driven suggestions!

Yours faithfully,

the authors

Reviewer 4 Report

Comments and Suggestions for Authors

1. What's "AI" in Abstract? Use the full name (artificial insemination) when it is appeared first time.

2. Use "cystic ovarian disease" instead of the abbreviation "COD" as the keyword; replace "ovarian" with "ovary" in the keywords.

3. Line 172: Do not start a sentence with a number like "37.2% (n=922) of the AI were performed on".

Author Response

Dear Reviewer,

First of all, we would like to thank you for your time and effort to improve the quality of our manuscript.

We tried to correct everything you suggested and was possible at this time. Below you can read and find the lines (L…) where the modifications have been made in the manuscript and the new manuscript file where you can see the yellow highlighted material including all the changes that were suggested by all 5 reviewers.

Original review

  1. What's "AI" in Abstract? Use the full name (artificial insemination) when it is appeared first time.

AU: Changed in the text (Line 44).

  1. Use "cystic ovarian disease" instead of the abbreviation "COD" as the keyword; replace "ovarian" with "ovary" in the keywords.

AU: Both have been changed as suggested.

  1. Line 172: Do not start a sentence with a number like "37.2% (n=922) of the AI were performed on".

AU: We changed these mistakes in the full text.

Thank you very much for your expertise-driven suggestions!

Yours faithfully,

the authors

Reviewer 5 Report

Comments and Suggestions for Authors

In a study by Vincze et al. the researchers attempted to examine the relationship between the type of corpus luteum (cavitary or compact) and ovarian cysts on the pregnancy rate and milk yield of Holstein-Friesian cows. The research was carried out on a large body of experimental and randomized material (more than a thousand cows), over an extended period of time (three years) under farm conditions. The purpose of the study is justified because until now it has not been possible to determine the effect of corpus luteum type on the reproductive performance in a large pool of Holstein cows.  The advantage of the study is the demonstration that 31-42 days after AI, the corpus luteum with a cavity, compared to the corpus luteum without a cavity, increases the risk that the cow will not be pregnant by 21 times.

Materials and methods: no comments

Results:

Line 196-208. I would suggest introducing a table or graph to make the manuscript more transparent, especially regarding pregnancy rate in the case of cavitary and compact CL, ovarian cyst and milk production.

Discussion

Line 217-218: It's hard to understand what the first part of the sentence means

Line 253-254: I wonder whether the limitation of your tests is the lack of ovarian assessment for the presence of cavitary CL before the first TRUS examination (e.g. between days 25 and 27)?

Line 256-258: please make it clearer, it's hard to understand the meaning of this sentence.

Line 285-286: Please explain whether and why COD is a problem for veterinarians, breeders and cattle owners?

The conclusions are consistent with the present evidence and arguments and they address the posed question.

The cited literature has been appropriately selected without any signs of self-citation.

The presented figures raise no doubts.

Author Response

Dear Reviewer,

First, we would like to thank you for your time and effort to improve the quality of our manuscript.

We tried to correct everything you suggested and was possible at this time. Below you can read and find the lines (L…) where the modifications have been made in the manuscript and the new manuscript file where you can see the yellow highlighted material including all the changes that were suggested by all 5 reviewers.

Original review

Materials and methods: no comments

Results:

Line 196-208. I would suggest introducing a table or graph to make the manuscript more transparent, especially regarding pregnancy rate in the case of cavitary and compact CL, ovarian cyst and milk production.

AU: A table has been inserted to make results clearer (Line 224).

Discussion

Line 217-218: It's hard to understand what the first part of the sentence means

AU: The sentence has been redefined. (Line 243-244).

Line 253-254: I wonder whether the limitation of your tests is the lack of ovarian assessment for the presence of cavitary CL before the first TRUS examination (e.g. between days 25 and 27)?

AU: Yes, as described above. We only could diagnose the type of the CL at pregnancy diagnosis.

Line 256-258: please make it clearer, it's hard to understand the meaning of this sentence.

AU: Line 283-285: sentence redefined:” However, the aim of the study was to evaluate those corpora lutea that remained cavi-tary during the early gestation and were associated with the positive or negative pregnancy status.”

Line 285-286: Please explain whether and why COD is a problem for veterinarians, breeders and cattle owners?

AU: Line 315-318. „With the intensive selection for milk production, COD became an everyday problem for bovine practitioners worldwide; hormonal disturbances, increased days open and in-creased veterinary costs cause significant economic losses associated with cystic ovarian disease.”

Thank you very much for your expertise-driven suggestions!

Yours faithfully,

the authors

Round 2

Reviewer 2 Report

Comments and Suggestions for Authors

I have no new suggestions

Author Response

Thank you very much for your valuable help in the improvement of our paper.

Reviewer 3 Report

Comments and Suggestions for Authors

Authors accomplished some of my suggestions and improved substantially the manuscript. However, I have some issues prior its publication:

1) Abstract still being very long as I previously commented. 

2) I would recommend using either corpora lutea or corpus luteum, or even CL... Please, take care when writting... line 68 for example...

3) I would deep more about the knowledge gap that supports the need of this study. Include it in the introduction section. Justification is completely important in these retrospective studies.

4) I still thinking that figure 2 is absolutely innecesary. I would merge figures 3 and 4 into one figure. Ultrasound images are nice, but we do all know about this.

5) Statistics: explain normality results. No correlations are done. No explanation about which variables from univariate you included in your GLMM.

6) Use numbers to explain results. If you would like to be grammatically correct and do not start a sentence with a number, please use words such as: "a total of...", or something similar, but avoit writting numbers with letters.

7) I think results should be improved. I already pointed it out. You need to maximize the results from your data. For example: missing DIM variable and its impact; missing COW as experimental unit, maybe cav CL persists over lactations, or even in the same lactation, or maybe dependent on number of AI; compare results by parity; etc. 

Try to maximize it all with the low sample size you are having (n=220)

I encourage authors to keep working on it. I think they did not take all the information from the data they collected. It could improve a lot the manuscript.

Comments on the Quality of English Language

minor

Author Response

Thank you for your valuable comments and suggestions on our manuscript.

Round 3

Reviewer 3 Report

Comments and Suggestions for Authors

Authors replied every suggestion. However: 

1) they still using corpora lutea and corpus luteum.

2) they included a short sentence to show the knowledge gap of their study. I encourage authors to improve this. It is the basis to undersand why these data and results are necessary... 

3) they included results due to "summer months" but it is not explained in M&M. Avoid using these terms, it is better to describe by THI. Summer it is not all the same in every country, nor hemisphere... 

4) shorten the discussion section. Moreover, my suggestion was included but as a result, you did not compare with another studies. Please improve it.

Comments on the Quality of English Language

minor

Author Response

Dear Reviewer,

Hereby we send you the line-by-line responses to your suggestions. Only the latest (3rd round) modifications are highlighted with yellow throughout the manuscript,

  • they still using corpora lutea and corpus luteum.

AU: I am not sure whether we correctly understood the problem with corpus luteum (when one structure is mentioned) and corpora lutea (when multiple structures mentioned), but we changed to the abbreviated version of both (CL and CLs throughout the text).

  • they included a short sentence to show the knowledge gap of their study. I encourage authors to improve this. It is the basis to understand why these data and results are necessary... 

AU: Following your recommendation, we inserted a longer explanation (Lines 60-66).

  • they included results due to "summer months" but it is not explained in M&M. Avoid using these terms, it is better to describe by THI. Summer it is not all the same in every country, nor hemisphere... 

AU: We have specified summer months (1st June to 31st August) Table, Line 273.

  • shorten the discussion section. Moreover, my suggestion was included but as a result, you did not compare with another studies. Please improve it.

AU: Following your suggestion, we have shortened the discussion section from 1444 words to 1043 words and inserted a short comparison with 2023 and 2024 studies (Lines 273-278).

Thank you very much for your valuable help and suggestions!

Yours faithfully,

the authors